# RAFT-Based Polymers for Click Reactions

**DOI:** 10.3390/polym14030570

**Published:** 2022-01-31

**Authors:** Elena V. Chernikova, Yaroslav V. Kudryavtsev

**Affiliations:** 1A.V. Topchiev Institute of Petrochemical Synthesis, Russian Academy of Sciences, Leninsky Prospect 29, 119991 Moscow, Russia; 2Faculty of Chemistry, M.V. Lomonosov Moscow State University, Leninskie Gory 1-3, 119991 Moscow, Russia

**Keywords:** click chemistry, azide–alkyne cycloaddition, thiol–ene and thiol–yne radical addition, Diels–Alder cycloaddition, RAFT polymerization, functional polymers, block copolymers, graft copolymers, hybrid polymers

## Abstract

The parallel development of reversible deactivation radical polymerization and click reaction concepts significantly enriches the toolbox of synthetic polymer chemistry. The synergistic effect of combining these approaches manifests itself in a growth of interest to the design of well-defined functional polymers and their controlled conjugation with biomolecules, drugs, and inorganic surfaces. In this review, we discuss the results obtained with reversible addition–fragmentation chain transfer (RAFT) polymerization and different types of click reactions on low- and high-molar-mass reactants. Our classification of literature sources is based on the typical structure of macromolecules produced by the RAFT technique. The review addresses click reactions, immediate or preceded by a modification of another type, on the leaving and stabilizing groups inherited by a growing macromolecule from the chain transfer agent, as well as on the side groups coming from monomers entering the polymerization process. Architecture and self-assembling properties of the resulting polymers are briefly discussed with regard to their potential functional applications, which include drug delivery, protein recognition, anti-fouling and anti-corrosion coatings, the compatibilization of polymer blends, the modification of fillers to increase their dispersibility in polymer matrices, etc.

## 1. Introduction

The non-trivial macromolecular architectures, including various chain topologies, block and graft copolymers, hybrid polymers, etc., have now become available due to accelerated progress in the techniques of anionic living polymerization and reversible deactivation polymerization [1,2]. The increasing interest in complex macromolecular structures is caused by their unique properties in solid state and solution, as well as by their intrinsic ability to self-assemble, making them highly promising in material applications. However, in some cases, the macromolecules composed of chemically different monomer units can hardly be synthesized through a single polymerization technique [3]. A similar problem often arises when grafting is used for the fabrication of hybrid structures [4]. The synthetic strategy depends on the number of factors including backbone, side-chain, and end-chain functionalities; the monomer unit sequence; chain topology; etc. If the switch from one polymerization technique to another requires multiple complex steps, then an easier solution comes from the coupling of the required building blocks into desired architecture through click chemistry.

The click chemistry concept was introduced by Sharpless et al. and, nowadays, it embraces a set of highly efficient chemical reactions [5,6,7,8]. Click reactions typically meet the following criteria: high efficiency and high yields, mild conditions, absence of by-products, stereospecificity, readily available reagents, and simple product purification techniques. The molecules with thiol, azide, alkynyl, alkenyl, and halogen groups are most commonly used in click chemistry. Some examples of click reactions are given in Figure 1. Azide–alkyne cycloaddition catalyzed by Cu(I) salt with organic ligands, one of the most popular click reactions, was used first in polymer chemistry to produce cyclic macromolecules and block copolymers [9,10,11,12,13,14,15]. However, it requires the purification of the products from copper catalyst, and that is why metal-free click reactions have become in demand. Among these reactions, thiol–ene and thiol–yne couplings, along with Diels–Alder and nitrile oxide–alkyne cycloadditions, are the most prominent [16,17].

Click reactions are not versatile and can only be conducted for certain polymers. Therefore, the design of a synthetic route is crucial to produce macromolecules with the desired architecture and functionalities. Since the 2000s, the numerous publications and reviews summarized the application of click reactions in the synthesis of macromolecules with complex architecture [16,17,18,19,20,21,22,23,24,25,26,27,28,29,30,31,32,33,34,35,36,37,38]. Nowadays, click reactions are used to assemble macromolecules, the parts of which are pre-prepared by living anionic polymerization, pseudo-living cationic polymerization, and reversible deactivation radical polymerization (RDRP) [39]. The latter provides the widest range of available functional groups in the monomer units. Among other RDRP techniques, reversible addition–fragmentation chain transfer (RAFT) polymerization is the most versatile and attractive due to its low sensitivity to the functional groups in monomers and solvents and mild conditions of realization [40].

The control over chain-end chemistry is possible if most of the chains are formed due to the RAFT agent (Figure 2). To meet this condition, one needs to use a RAFT agent that is effective in the polymerization of chosen monomers, to avoid side reactions (such as intermediate radical termination), and to choose an appropriate ratio of the RAFT agent to initiator ([RAFT agent]/[initiator] >> 1). Nowadays, hundreds of RAFT agents with various types of R and Z groups have been described and applied to the controlled synthesis of (co)polymers [41,42,43,44]. Besides, the approaches for quantitative modification of the thiocarbonylthio group, resulting in its conversion into –SH, –CH=CXY, –CH_2_–CHXY, –R′ (product of the free radical initiator decomposition), –O–NR_1_R_2_, etc., have been developed [45,46,47]. Thus, one can readily synthesize a polymer with a given type of end groups, which may be converted to other functional groups on demand.

The strategy of combining RAFT polymerization with click chemistry should depend on the desired architecture and type of functional groups (Figure 3). R and Z groups play different roles in the RAFT mechanism [48]. The Z group is known as a stabilizing group; its structure, i.e., alkyl, aryl (dithioesters), O–alkyl, O–aryl (xanthates), N–alkyl, N–aryl (dithiocarbamates) or S–alkyl, and S–aryl (trithiocarbonates), affects the reactivity of the C=S bond and the stability of intermediate radicals. Thus, the Z group participates in multiple steps of reversible chain transfer reactions with propagating radicals. Hence, the Z group should keep its stability (functionality) throughout polymerization. In contrast, the R group, known as a leaving group, undergoes the fragmentation step after the formation of an intermediate radical and participates in a single step of reinitiation via the reaction with a monomer. As a result, the choice to functionalize R and Z groups with clickable substituents, either in an initial RAFT agent or a resulting polymer, depends on the nature of a monomer and polymerization conditions.

In special cases, RAFT agents containing clickable R groups are developed. Here, RAFT polymerization with production of the “building blocks” is followed by the corresponding click reaction (route *a*) that leads to various block copolymers or grafting to a surface, nanoparticle, biomolecule, etc. More often, RAFT agents contain a functional group R that can be converted to a clickable group (route *b*); this modification is required before the click reaction. Another possibility is the modification of a ZC(=S)S group followed by the click reaction (route *c*). A clickable group at only one chain end gives rise to the synthesis of various block copolymers, grafting to different objects, or, in some cases, to the synthesis of star macromolecules. The presence of clickable groups at both chain ends (route *a*/*b* + route *c*) allows the synthesis of cyclic macromolecules. An interesting approach deals with the preliminary modification of side groups followed by the click reaction (route *d*). In this case, various branched structures, such as the graft copolymers, brushes, etc., are expected.

In this review, we discuss the implementation of the above strategies to the controlled synthesis of complex macromolecular architectures and hybrid polymer systems. As far as we are aware, this is the first attempt to analyze this problem based on the position and type of functional groups in polymer chains. We will address the possible ways of involving known RAFT agents (dithioesters, trithiocarbonates, xanthates, and dithiocarbamates) in click reactions, either directly or via the modification of R or ZC(=S)S groups. The distinctive features of the corresponding click reactions of chain-end groups will be discussed. Special attention will be paid to the post-modification of the side groups of macromolecules for creating clickable groups and to the formation of various grafted structures. 

## 2. R/Z Group Approach

A clickable RAFT agent should contain one of the suitable functional groups, e.g., −N_3_, −C≡C, −C=C, −N=C=O, etc. Their locus in the leaving group R seems preferable from the viewpoint of the RAFT agent synthesis. In contrast, the presence of −SH, which is also attractive for clicking in the initial RAFT agent structure, seems inappropriate because of the high probability of thiol participation in conventional chain transfer reactions that affects the RAFT polymerization mechanism.

The copper-catalyzed azide–alkyne cycloaddition is routinely used for the RAFT-based polymer end-coupling. In that regard, a number of azide, alkyne, and trimethylsilyl-protected alkyne functional RAFT agents have been described in the literature (Table 1). Their application results in the end-functionalized polymers, e.g., polystyrene (PS) and poly(vinyl acetate) (PVAc) [49,50], as well as PVAc and glycopolymer [51], which can be clicked together to produce block copolymers or grafted onto inorganic or organic nanoparticles [52,53,54,55]. Since both protected and unprotected alkyne groups keep their functionality in the course of radical polymerization, protection of the alkyne substituent makes little sense. 

Among different polymers, PS, polyacrylates, and polyacrylamides are the most common [49,50,51,52,53,54,55,56,57,58,59,60,61,62,63,64,65,66,67,68,69,70,71,72,73,74,75,76,77]. Sometimes, methacrylic monomers are involved in the combination of RAFT polymerization with click chemistry [78,79]. However, amine-containing monomers (e.g., vinyl pyridine) should be used with caution because of the possible complexation with copper ions catalyzing the azide–alkyne interaction.

It is noteworthy that the azide–alkyne cycloaddition is more popular for the stabilization of nanoparticles with grafted polymers than for the block copolymer synthesis, which has a natural explanation. Indeed, if a click reaction proceeds by end-coupling, the concentration of the functionalized groups becomes very important. In the examples given above, the concentration of polymer chains is rather high (0.01–0.02 mol⋅L^−1^), whereas the polymer molar masses (MM) are 2–5 kg⋅mol^−1^ only. Even in this situation, the MM distributions of the click products usually reveal tails that can be assigned to the initial functionalized polymers, thus indicating the incompleteness of the reaction.

**Table 1 polymers-14-00570-t001:** Azide- and alkyne-functionalized RAFT agents.

Type	Z Group	R Group	Ref.
Dithioester	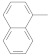	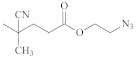	[56]
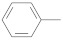	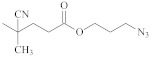	[49,57,59]
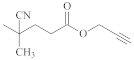	[49,59]
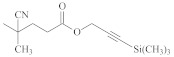	[49,51,60]
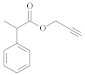	[54]
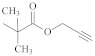	[58,66,67]
Trithiocarbonate	C_12_H_25_S–	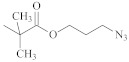	[57,61,62,63]
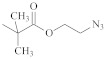	[64,65]
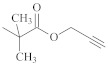	[55,68,69,70,71]
CH_3_S–	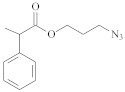	[52,53,74]
C_3_H_7_S–	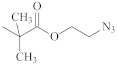	[72]
*i*C_4_H_9_S–	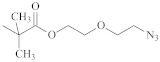	[73]
Xanthate	C_2_H_5_O–	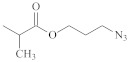	[75,76]
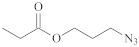	[49,51,60]
N_3_CH_2_CH_2_O–	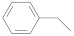	[77]
Dithiocarbamate	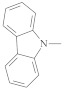	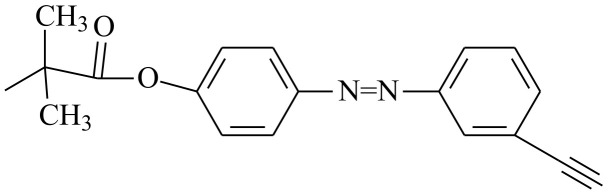	[50]

An increase in the polymer MM makes its dissolution at such high concentrations impossible. At the same time, the accessible concentrations of the end-functionalized polymers with MM higher than 10 kg⋅mol^−1^ lead to slow rates and low conversions of the click reaction in solution. Another situation occurs, when the functionalized polymer chains are confined in the aggregates or micelles [54] or adsorbed at the surface of nanoparticles [55]. The average concentration of the clickable groups is low, but their local concentration is high, speeding up the reaction. Moreover, easy functionalization of the surface of inorganic nanoparticles makes this approach rather attractive. 

In some applications, the products of the copper-catalyzed azide–alkyne cycloaddition may require purification from the catalyst. Therefore, the metal-free approaches known as ‘additive-free clicking’ [80] are also in demand. They include thiol–ene and thiol–yne couplings, thiol–isocyanate addition, pyridyl–disulfide exchange, Diels–Alder, and hetero Diels–Alder reactions [17,18,19,20,21,22]. Thiol-based systems are often insufficient to provide complete polymer–polymer coupling and prone to the formation of disulfides, especially in oxophilic aqueous solvents [81]. 

Among additive-free clicking reactions, a tetrazine–norbornene inverse-electron-demand Diels–Alder conjugation (DA_inv_) was successfully applied to polymer–polymer coupling [80]. The reaction between tetrazines and strained alkenes or acetylenes yields dihydropyridazines or pyradizines (Figure 4). It is fast, atom-efficient, catalyst-free, air-insensitive, and quantitative so that all requirements of the click concept [82] are fulfilled.

A number of norbornenyl-terminated polymers, e.g., PS and poly(N-isopropyl acrylamide), were synthesized via RAFT polymerization using the norbornenyl-functionalized RAFT agent, 4-({[2-*exo*]-bicyclo[2.2.1]hept-5-en-2-ylmethoxy} methyl)benzyl butyl carbonotrithioate (**I**) presented in Figure 1, and coupled with tetrazine-terminated polymers, poly(d-valerolactone), and poly(ethylene glycol).

However, even in that case, the polymer MM is limited since the reaction between the terminal groups sets an upper limit on the available concentrations of reactants.

Another popular click reaction involves 1,2,4-triazoline-3,5-dione and diene (Figure 5). This additive-free Diels−Alder ene-type coupling proceeds with a super fast rate at ambient conditions [83].

The triazolinedione-terminated poly(butyl acrylate) was synthesized using trithiocarbonate **II** (Figure 2) and successfully coupled with poly(ethylene glycol) terminated by diene. The reaction was completed in 5 s at room temperature, being the most rapid click reaction in polymer chemistry to date [83].

Sulfur(VI) fluoride exchange (SuFEx) reaction, identified as orthogonal to RAFT radical polymerization, is a promising candidate for successful click coupling (Figure 6) [84]. Indeed, both reactive groups are stable toward radicals, and they do not interact with double C=C bonds of vinyl groups. This reaction has already demonstrated its potential in polycondensation and side chain modification [85,86,87,88]. Only recently, the SuFEx chemistry was applied to block copolymer synthesis with the assistance of newly developed RAFT agents **III** and **IV** (Figure 3) [89,90]. These RAFT agents are versatile to the polymerization of a number of monomers, e.g., methyl methacrylate, styrene, tert-butyl acrylate, and 4-acrylomorpholine. They provide the formation of block copolymers through the coupling of fluorosulfonyl benzyl with ((trialkylsilil)oxy)phenyl groups.

Besides leaving the clickable R group on a macromolecule, the thiocarbonylthio ZC(=S)S group can also be utilized in hetero Diels–Alder coupling reactions. When provided with a suitable Z group functionality, the C=S bond is able to undergo cycloaddition with dienes (Figure 7) [91]. RAFT-based PS containing the Z group depicted in Figure 7 was coupled with polycaprolactone in chloroform solution in the presence of zinc chloride at 50 °C. A complete conversion was found after a reaction time of 24 h. Nevertheless, similarly to the above approaches of polymer coupling, a high concentration of reactants was required (~0.03 mol⋅L^−1^) to provide a satisfactory rate of the reaction, while the reaction product contained traces of the initial unreacted polymers. 

Significant progress in the macromolecular engineering of click reactions was achieved in [92], where the block copolymers of PS and poly(isobornyl acrylate) with MM ranging from 34 to over 100 kg⋅mol^−1^ were obtained. To that end, cyclopentadienyl end-functionalized PS and poly(isobornyl acrylate) pyridin-2-yldithioformate were coupled in chloroform for 10 min for low-molar-mass polymers and for 2 h for high-molar-mass polymers (Figure 8). A noticeable increase in MM of the initial polymers demanded high dilution of the reaction media. However, the reaction efficiency was extremely high, since no traces of the initial reactants were discovered by size exclusion chromatography (SEC) and UV-spectroscopy even for the polymers with the highest MM.

Further development of this approach led to the creation of a dual functional RAFT agent (**V**) depicted in Figure 4 [93]. In combination with photoenol chemistry, a triblock copolymer of poly(isoprene-co-styrene)-*b*-(poly(ethyl acrylate)-*b*-(poly(ethylene oxide) was synthesized through the thermally induced hetero Diels–Alder cycloaddition followed by the photo-induced ligation coupling of three different macromolecules into one chain.

Another methodology for the synthesis of α-functional polymers in a one-pot simultaneous polymerization accompanied by an isocyanate “click” reaction was described in [94]. It is based on the preparation of a carbonyl-azide RAFT agent precursor that undergoes in situ Curtius re-arrangement during RAFT polymerization, yielding well-controlled polymers modified with α-isocyanate (Figure 9). Their coupling with polymers bearing a terminal OH group may easily result in the formation of block copolymers.

This strategy overcomes numerous difficulties associated with the synthesis of a polymerization mediator bearing an isocyanate moiety in the R group and with handling of this highly reactive functional group. The new carbonyl-azide RAFT agent is effective in the polymerization of (meth)acrylates, acrylamides, and styrenes providing end-functional polymers with narrow MMD.

A rather important type of RAFT agents shown in Figure 5, applicable for bioconjugation, bears a dithiopyridine group (**VI**–**VIII**). This group is kept stable throughout radical polymerization and undergoes exchange with a desired thiolated (macro)molecule under mild conditions [95,96,97,98,99,100,101].

The versatile approach to the synthesis of clickable RAFT agents is proposed in [102], which is based on the reaction of thiolactone and amine, resulting in the formation of trithiocarbonates with desired functionality. Some examples of the functional groups, as well as the synthetic strategy, are given in Figure 10.

Finally, it should be mentioned that clickable RAFT agents may be readily produced through the modification of commercially available RAFT agents. Typical examples shown in Figure 11 are the reactions of the carboxyl moiety of a RAFT agent or polymeric RAFT agent with 3-azido-1-propanol (incorporation of azide functionality [103]) and with propargyl alcohol, optionally protected with the trimethylsylil group (incorporation of alkyne functionality [49,79]).

Thus, the diverse chemistry of RAFT agents provides a wide range of functionalities in R or Z groups which are suitable for click reactions and stable throughout the polymerization process.

## 3. ZC(=S)S Group Modification Approach

A macromolecule formed through the RAFT mechanism contains a thiocarbonylthio ZC(=S)S group on its ω end. The amount of these chains is determined by efficiency of the RAFT agent and the molar ratio of the RAFT agent to initiator decomposed by the time that the polymerization is completed [40]. The ZC(=S)S group is rather convenient for further modifications, e.g., reactions with nucleophiles resulting in thiol formation (*a*), thermolysis or radical-induced oxidation providing C=C bond formation at the chain end (*b*), addition–fragmentation coupling with the radical initiator (*c*), etc. (Figure 12) [45,46,47]. Some products of the above reactions may be considered as precursors for click reactions. We will address these cases below and classify them by the chemical nature of the stabilizing group Z in the RAFT agent.

### 3.1. Conversion of the ZC(=S)S Group to the Thiol Group

RAFT-based polymers are a versatile source of macromolecules with chain-end thiols, which are suitable for various thiol–X click reactions (X = -ene, -yne, Hal, etc.) and for further modification of chain-end functionality (Figure 13) [41,42,43,44,45,46,47]. 

The conversion of thiocarbonylthio moiety in the ZC(=S)S group occurs through reactions with appropriate nucleophiles [104,105,106,107,108,109,110,111,112,113,114,115,116,117,118,119,120,121,122,123,124,125,126] (Table 2) that include mainly primary amines [105,106,107,108,109,110,111,112,113,114] and hydrazine (or hydrazine hydrate) [116,117,118,119,120,121], alkali [122], and NaN_3_ [115], along with reducing agents such as NaBH_4_ [123,124,125,126,127] and LiB(C_2_H_5_)_3_H [128]. The choice of the nucleophile depends on the chemical nature of the stabilizing group Z. If a suitable nucleophile or hydride is chosen, then the reaction proceeds rather rapidly with full conversion, but yields disulfide as a typical by-product. Thus, it is desirable to perform these reactions under oxygen-free conditions and in the presence of a reducing agent, such as tributylphosphine or tris(2-carboxyethyl)phosphine [127], in order to suppress the formation of disulfide. In some cases, a two-stage process is preferable, which includes the initial reaction with nucleophile without regard for the disulfide formation and the following treatment with a reducing agent to obtain a pure thiol. The completeness of the reaction can be easily controlled with UV spectroscopy and SEC methods.

The nature of the monomer unit neighboring with ZC(=S)– substituent of the Z group and of nucleophile or reducing agent may influence the reaction kinetics and the yield of a polymeric thiol. For example, PS and PMMA with dithiobenzoate end groups undergo different pathways in aminolysis [111], the end groups being mostly converted to disulfide and thiolactone groups, respectively. PDMAEMA and PLMA reveal the behaviors similar to PMMA. If trithiocarbonate (Z = SC_12_H_25_) is used instead of dithiobenzoate in the RAFT polymerization of styrene, then the aminolysis will result in the formation of both polymeric thiol and disulfide [146]. The efficiency of NaBH_4_ and primary amines in the conversion of dithiobenzoate and trithiocarbonate terminal groups of PMMA into thiols is different [124]. NaBH_4_ works better for dithiobenzoate, whereas amines are more appropriate for trithiocarbonate. In both cases, disulfide moieties are absent.

With the use of NaN_3_, deoxygenation can be avoided [115]. Polymeric dithioesters (Z = phenyl or naphtyl) and xanthate readily convert into disulfides, regardless of the nature of the polymeric substituent (PMMA, PS, PBA, or PVAc).

Dithiocarbamate chain-ends are substantially less susceptible to a nucleophilic attack than other RAFT ZC(=S)S groups (dithioesters, trithiocarbonates, and xanthates) [153,154]. The experimental results are confirmed by theoretically estimating the order of reactivity of Z groups toward primary amine nucleophiles. The activity of RAFT agents falls in the series: dithioate > dithiobenzoate > xanthate > trithiocarbonate >> dithiocarbamate [154]. The unusual behavior of dithiocarbamate-terminated dimethyl acrylamide was observed during the aqueous size exclusion chromatography in the presence of a low amount of NaN_3_ [155]. Partial end-group loss has occurred, resulting in the polymer with a thiol end.

Finally, a simple way to produce new thiols based on the exchange reaction between dithioesters and thiols should be mentioned [156]. This approach is also used in the synthesis of low-molar-mass RAFT agents [157,158]. Polymeric thiols have been recently derived from the dithiobenzoate-terminated polymers through their thiol–thioester exchange with cysteine [159,160] or homocysteine [160].

As opposed to other polymeric RAFT agents, xanthate-based polymers can be transformed to polymeric thiols by simple heating either in bulk or solution [161,162]. At high temperature, which depends on the nature of a leaving R group in polymeric xanthates (Table 3), the Chugaev elimination occurs producing a polymeric thiol and alkene (Figure 14) [161]. This mechanism holds independently of the nature of the monomer unit bound to the xanthate group (styrene, *t*-butyl acrylate, or dimethyl siloxane).

This methodology was used in RAFT polymerization of NIPAM mediated by thiolactone-based xanthate [112]. Under the thermolysis conditions, the synthesized polymer converts to a polymeric thiol through the Chugaev elimination. However, the terminal thiol group is unstable and undergoes thiolactonization with the penultimate NIPAM unit. The same idea was realized in [163] through RAFT polymerization of dimethyl vinylphosphonate mediated by xanthate, followed by the thermolysis of the resulting oligomer and thiol–ene click reaction with styrene-butadiene rubber.

To summarize, a number of suitable methods to convert thiocarbonylthio end groups into thiol groups have been developed. The better results are achieved when polymeric thiols are immediately involved in a thiol–X click reaction, resulting in the polymers with a wide range of chain-end functionalities, various bioconjugates, and polymer-stabilized nanocomposites [21,25,28,30,112,114,133,141,143,148,149,150,151,152,164]. 

### 3.2. Conversion of the ZC(=S)S Group to the C=C Group

The thermolysis of RAFT-based polymers helps to remove the ZC(=S) group and produce polymers with the C=C end group (except xanthates, see 3.1) [41,42,43,44,45,46,47]. Thermal stability of RAFT-based polymers, mechanism of thermolysis, and purity of the product depend on the nature of the terminal monomer unit and thiocarbonylthio species [165,166,167,168,169,170,171,172,173,174,175,176,177,178]. In general, the thermal stability of RAFT-based polymers decreases in the series poly(vinyl acetate) ~ poly(N-vinyl pyrrolidone) > poly(butyl acrylate) > poly(N-isopropylacrylamide) > polystyrene > poly(methyl methacrylate), and among the Z groups as dithiobenzoate > trithiocarbonate > xanthate ~ dithiocarbamate [165,167,168,169,170,171,177]. Thus, the polymers formed by “less activated monomers” have higher thermal stability than the polymers formed by “more activated monomers”. Low-molar-mass RAFT agents are typically less stable than RAFT-based polymers [178]; however, the study by Zhou et al. doubts this statement [173]. According to [173], the stability of the dithiobenzoates decreases in the order CH_2_Ph > polystyryl > CHPh(CH_3_) > C(CH_3_)_2_COOC_2_H_5_ > CPh(CH_3_)_2_ > PMMA > C(CN)(CH_3_)_2_. The authors believe that the reason for this phenomenon originates from various mechanisms of the decomposition of RAFT agents with different leaving groups.

The chemical nature of a polymeric substituent (Table 4) and stabilizing Z group (Table 5) determines the kinetics and mechanism of the thermal degradation of RAFT-synthesized polymers. For example, poly(butyl acrylate) with butyl trithiocarbonate group undergoes C–S bond homolysis, backbiting, and β-scission, which result in the formation of a macromonomer with an exo-methylene double bond [165,167]. The thermolysis of polystyrene containing the same terminal group proceeds via the Chugaev elimination mechanism, being accompanied by the elimination of a butyl trithiocarbonate group and the formation of a PS macromonomer [165]. In the case of a xanthate-terminated PS, the polymer with a thiol end group is formed [161]. Thermal stability of the polymers synthesized using acid-base “switchable” dithiocarbamates is strongly dependent on the identity of the polymeric substituent and on the state of the “switchable” pyridyl group (Table 5) [161]. 

The multiple TGA data reveal some discrepancy with the thermolysis performed under isothermal conditions, with RAFT-based PMMA being the typical example. According to the literature, PMMA with a methyl trithiocarbonate group undergoes three stages of mass loss with the first step starting at 170 °C [158]. A three-stage degradation process was described for PMMA with a dodecyl trithiocarbonate end group, with the first step starting at ca. 160 °C and the activation energy being about 185 kJ⋅mol^−^^1^ [180], whereas a three-stage degradation with the first step at ca. 100 °C was reported in [177]. A three-stage degradation process starting at 120–130 °C was detected for the dithiobenzoate-terminated PMMA [134], whereas a two-stage degradation process starting in the similar temperature range was reported in [177] and, finally, a single-stage thermal degradation above 200 °C was described in [158]. The study of this process under isothermal conditions has not made the situation any clearer. The cleavage rate for both dithiobenzoate and methyl trithiocarbonate end groups at 180 °C was comparable for PMMA with *M*_n_ = 10–50 kg⋅mol^−^^1^ [167]. The authors of [173] reported that the dithiobenzoate group cleavage in PMMA takes place at a rate of 4.05⋅10^−^^5^ s^−^^1^ at 120 °C with the activation energy of 77.6 kJ⋅mol^−^^1^, irrespective of the polymer MM. A loss of PMMA chain-end functionality clearly manifests itself upon 24 h of the isothermal treatment at 120 °C for dithiobenzoate end groups and already at 100 °C for trithiocarbonate ones [177]. At 120 °C (for trithiocarbonate) and 140 °C (for dithiobenzoate), the ability of both polymers to serve as macromolecular RAFT agents in the chain extension processes ceases almost completely. In the case of dithiobenzoate groups, the thermolysis reaction is of the first order, whereas trithiocarbonate groups containing two C−S bonds demonstrate more complex degradation kinetics.

The mechanism of dithiobenzoate group cleavage in polymers is considered similar to that of the Chugaev reaction (Figure 15a) [181]. Usually, the thermolysis of dithiobenzoate-based polymers yields polymeric products with the molar mass and dispersity close to the characteristics of the parent polymer. A questionable issue is the mechanism of cleavage of the trithiocarbonate end group. Without doubt, it strongly depends on the nature of the polymeric substituent, but the type of the low-molar-mass substituent R’ in the stabilizing group –SR’ (Figure 15b–d) can also be important. The elimination mechanism shown in Figure 15b is typical for PS with a trithiocarbonate end- or mid-chain group [168,169,170]. Homolysis followed by backbiting and chain scission is typical for polyacrylates [168,174]. For PMMA with a trithiocarbonate end group, in some cases, a drop in MM is observed, and the end group removal is believed to occur via the homolysis of C–S bond followed by depolymerization, a process that continues until radical annihilation (Figure 15d). According to other data, the MM can be kept the same, while the kinetics of the trithiocarbonate group loss does not follow the first order, so that a more complicated mechanism of consecutive elimination of trithiocarbonate group is proposed [177]. Another possibility to replace the ZC(=S)S group with a C=C one is the addition of a small (ppm) amount of Co(II) compound which plays the role of a catalytic chain transfer agent during the later stages of RAFT polymerization [182]. However, this approach, which may be considered as the radical-induced oxidation (Figure 16), is applicable to methacrylate monomers only.

To summarize, the thermolysis of RAFT-based polymers in many cases results in the formation of pure polymers containing C=C terminal units. Such a type of polymers may be involved in thiol–ene click reactions.

### 3.3. Reaction of Addition–Fragmentation Coupling of the ZC(=S)S Group with Radical Initiator

The ability of RAFT-based polymers to participate in addition–fragmentation reactions with radical species underlies the idea to replace the ZC(=S)S group with another functional group. This method implemented for the first time by Perrier et al. [183] involves heating a RAFT-synthesized polymer with a large excess (typically ∼20 molar equivalent to ZC(=S)S groups) of a radical initiator.

The general mechanism of this process is given in Figure 17. Most often, azo initiators, such as AIBN, are used [47]. In the case of dithiobenzoate RAFT agents, this method is most effective for the polymers with a terminal methacrylate unit [41]. For styrene and acrylates, a very large excess of initiator is required for completeness of the reaction [170]. According to Figure 17, the radical X^•^ formed upon the decomposition of the radical initiator reacts at the C=S bond of the thiocarbonylthio group, forming an intermediate carbon-centered radical Int1 (1). Its fragmentation can either release the attacking radical X^•^ or liberate the polymer radical P_n_^•^. At an essentially high concentration of radicals X^•^, the equilibrium shifts to the right, even in the absence of monomers. The better leaving P_n_^•^ group suggests that the smaller excess of the initiator is required to shift the equilibrium. The released polymer radical P_n_^•^ reacts with the excess of the initiator radicals X^•^, forming the substitution product P_n_–X (2). The low-molar-mass products formed through the reactions (3), (4), (8), and (9) can easily be separated from the polymer by precipitation. The reaction (5) is typically avoided in the case of the large excess of initiator, whereas the reactions (6) and (7) play a minor role at low concentrations of the initial P_n_SC(=S)Z. Apart from azo-initiators, peroxides (lauroyl-, benzoyl peroxide) [184,185,186] or their mixtures (2 equivalents) with azo-initiators (20 equivalents) can also be used [185]. However, in this case, the task is to remove the ZC(=S)S group from the chain end rather than to provide it with a new functionality.

Figure 17 is valid for dithioesters, nonsymmetrical trithiocarbonates, and dithiocarbamates, but for symmetrical trithiocarbonates, it is supplemented with additional reactions [170,187]. It should be noted that, in the latter case, MM of the polymers involved in the reaction with an excess of radical initiator decreases by about half. AIBN is the azo-initiator most popular for the removal of thiocarbonylthio groups from the mentioned classes of RAFT agents. Some examples of its application are given in Table 6. 

Among the azo-initiators used in addition–fragmentation–coupling reactions, 4,4’-azobis(4-cyanopentanoic acid) [127,132,152,153]; 2,2’-azobis(5-hydroxy-2- methylpentanenitrile) [132]; and **IX** [139,140], **X** [154,155], and **XI** [149] compounds depicted in Figure 6 are attractive from the viewpoint of further application of chain-end modified polymers in click reactions.

Nowadays, this type of end-group transformation is also used to create a desired functionality or to broaden applications of known polymers. For example, Bohec et al. [195] combined removing the trithiocarbonate end groups, achieved in alkyne-R-group-functionalized PNIPAM by the reaction with AIBN, with a thiol–yne click reaction. RAFT copolymerization of N-acryloylmorpholine and N-succinimidyl acrylate was carried out using a dithiobenzoate, bearing a hydroxyl substituent in leaving R group [196].

The dithiobenzoate moiety was replaced using VA-086 azo-initiator containing the OHCH_2_CH_2_NHC(=O)(CH_3_)_2_C group. Upon acryloylation of the hydroxyl terminal groups, a telechelic with C=C bond at both chain ends was formed. This polymer may be considered as a macromonomer for thiol–ene click reaction with proteins via the conjugation of pendant succinimidyl esters. A similar approach was applied to produce (1) telechelic homo and block copolymers of PMMA, poly(diethylene glycol monomethyl ether methacrylate), poly(ethylene glycol monomethyl ether methacrylate), and poly(lauryl methacrylate) bearing terminal pentafluorophenyl cyanovaleriate groups able to participate in the click reaction with thiols and amines [197]; and (2) hetero telechelic biotin–maleimide PNIPAM able to form streptavidin–bovine serum albumin polymer conjugates [198]. Zhang et al. used AIBN for the cleavage of the trithiocarbonate end group in PNIPAM followed by ring-opening polymerization of γ-benzyl-L-glutamate N-carboxyanhydride that led to a double thermo- and pH-sensitive diblock copolymer [191]. An interesting type of the ZC(=S)S group modification into a alkyne group (Figure 18) was suggested in [199]. It enabled the click reaction between propargyl acrylate and dithiobenzoate groups that yielded a block copolymer-based prodrug.

## 4. R Group Modification Approach

RAFT-based polymers with latent R group can also be modified to produce clickable polymers. The approaches for this purpose are, in principle, similar to those described in Section 2. The modification reactions must be efficient and must not affect the side-chain functionalities. 

The typical examples of incorporating the terminal functional groups for azide–alkyne cycloaddition are shown above in Figure 11. The presence of –COOH moiety in the structure of R group located at the α end of the macromolecule enables its reaction with 3-azido-1-propanol, leading to N_3_ functionality [102], and with propargyl alcohol or trimethylsylil propargyl alcohol, resulting in C≡C- or (CH_3_)_3_Si–C≡C functionality [49,79]. The reactions are conducted in common organic solvents at a low temperature in five-fold excess of the modifying reactant. 

Azide functionality can also be introduced through the modification of a perfluorobenzene substituent in the R group of the commercially available RAFT agent by treatment of the polymer with sodium azide, similar to the procedure described in [35] for perfluorobenzyl methacrylate (Figure 19).

An OH moiety in the structure of the R group located at the α end of the macromolecule may be converted subsequently to Br- and cyclopentadienyl functionalities, with the latter being able to participate in the Diels–Alder click reaction (Figure 20) [92].

Urazol substituent of a RAFT-based polymer may easily be converted to triazolinedione that is eager to react with a diene-functionalized polymer (Figure 21) [83].

In general, the modification of the R group may be performed in both initial and polymeric RAFT agents. The choice depends on the stability of the required functionality under polymerization conditions. If it is low, then RAFT polymerization followed by post-modification is preferable. Otherwise, the synthesis of a low-molar-mass RAFT agent with the desired functionality of R group is possible.

## 5. Post-Modification of Side-Group Approach

The abundance of monomers that can be involved in RAFT polymerization provides broad opportunities for the post-modification of polymer side groups. Independent of the type of reactants, such modification often leads to branched architectures, which can complicate the attainment of high yields and product purification. Therefore, even if a formal click reaction is carried out, in each case, it is necessary to make sure that the product is a well-defined polymer.

As discussed above, copper-catalyzed azide–alkyne cycloaddition (the Huisgen reaction) is chosen most frequently. In principle, triazole rings, produced by this reaction and known for their ability to bind metal ions and antibacterial activity, can be synthesized by the same azide–alkyne interaction in the monomeric form and then polymerized by RAFT for 8 h to attain a decent values of *M*_n_ = 10 kg⋅mol^−^^1^ and *Ð* = 1.4 at 69% conversion [200]. In that study, the azide monomers contained naphthalene rings, which became polymer pendants able to form complexes with 5 to 6 Sm(III) ions per macromolecule. Yan et al. [201] did essentially the same but with an ethynylcobaltocenium hexafluorophosphate monomer (Figure 22). The azide–alkyne click was also followed by RAFT polymerization with a conversion of 50% due to the radical termination. The product was characterized as a cationic cobaltocenium-containing polyelectrolyte with redox characteristics. In a study by Wu et al. [202], triazole units were formed between the azide side groups of a random copolymer of 2-(trimethylsilyloxy)ethyl methacrylate and methyl methacrylate (*M*_n_ = 19 kg⋅mol^−^^1^ and *Ð* = 1.22), synthesized by atom transfer radical polymerization and subsequently subjected to desilylation in THF; esterification with 2-bromoisobutyl bromide; and treatment with sodium azide and an alkyne-terminated RAFT agent known as S-1-dodecyl-Sʹ-(α,αʹ-dimethyl-αʹʹ-propargyl acetate) trithiocarbonate. The modified polymer was not that narrow-dispersed (*M*_n_ = 27 kg⋅mol^−^^1^ and *Ð* = 1.50), yet it was suitable for grafting RAFT polymerization of N-isopropyl acrylamide into side chains and yielding a macromolecular brush, which demonstrated self-assembly behavior in aqueous solutions.

Most studies combining RAFT and a click concept start with the polymerization step. If two complementary reactive groups belong to a same chain, the subsequent click reaction between them leads to a nonlinear macromolecule. Shi et al. [58] synthesized poly(*N*-isopropylacrylamide)-*b*-polystyrene diblock copolymer with an acetylene-terminated styrene block and azide group at the junction between two blocks. An intramolecular click reaction performed at high dilution resulted in a tadpole polymer with a PS cycle and poly(*N*-isopropylacrylamide) tail (Figure 23), which possessed more pronounced amphiphilic properties compared with its linear analogue.

In many studies, a click reaction was used for the modification of polymer side groups with low-molar-mass compounds or oligomers. Y. Li et al. [203] polymerized 2-azidoethyl methacrylate in the presence of α-cyanobenzyl dithionaphthalate as a RAFT agent that provided excellent control over the MMD at 30–40 °C up to high conversions (*Ð* < 1.15 for *M*_n_ = 50 kg⋅mol^−^^1^). Then, the polymer with pendant azide moieties was either clicked with a slight excess of phenyl acetylene or used as a macro RAFT agent for the block copolymerization with methyl methacrylate, while keeping a narrow MMD. G. Li et al. [204] synthesized a new monomer, 2-chlorallyl azide, and copolymerized it with methyl acrylate at room temperature in the presence of benzyl 1H-imidazole-1-carbodithioate to obtain a polymer with *M*_n_ ~ 10 kg⋅mol^−^^1^ and *Ð* < 1.21 containing 3–6 pendant azide groups per chain. The resulting copolymer could be either extended with 2,2,3,4,4,4-hexafluorobutyl acrylate block up to *M*_n_ ~ 20 kg⋅mol^−^^1^, keeping a narrow MMD and intact azide groups, or clicked together with alkyne-capped poly(ethylene glycol). In the latter case, the graft product was shown to form a hydrophobic coating on a glass surface upon azide photocrosslinking with UV light. Ebbesen et al. [205] demonstrated the potential of acrylic polymers with azide side groups for bioconjugation. They synthesized N-(3-azidopropyl)methacrylamide and copolymerized it with N-(2-hydroxypropyl)methacrylamide in the presence of 4-cyanopentanoic acid dithiobenzoate as a RAFT chain to obtain a copolymer with *M*_n_ = (10–54) kg⋅mol^−^^1^ and *Ð* = 1.02–1.06 that contained up to 5 mol.% of azide-functionalized monomer units. Then, it was conjugated by the Huisgen reaction with different biorelated alkyne-containing substances, Atto 647N propargyl near-infrared dye (with azide conversion of 58%), phosphocholine (conv. 87–88%), and poly(ethylene glycol) with different substitution degrees (conv. 80–92%). These results open new opportunities for the use of poly(N-(2-hydroxypropyl)methacrylamide) in drug delivery. Tian et al. [206] prepared a Janus macromolecular brush using both side functionalities, azide- and hydroxyl-, formed upon opening oxirane rings in the glycidyl methacrylate units of poly(ethylene glycol)-*b*-poly(glycidyl methacrylate) block copolymer synthesized by two-step RAFT polymerization. Azide groups were clicked together with alkynyl-functionalized tetraphenyl porphyrins followed by the esterification of hydroxyl groups with 4-cyano-4- (ethylsulfanylthiocarbonyl)sulfanyl pentanoic acid used at the next stage as a RAFT agent for the polymerization of pH-responsive 2-(diisopropylamino)ethyl methacrylate. Such brushes are expected to find application in photodynamic therapy as photosensitizers and in chemotherapy as pH-responsive vehicles for the release of antitumor drugs.

Intermacromolecular click reactions usually aim at the synthesis of graft copolymers or grafted surfaces with desired functionalities. For the Huisgen reaction, both alkyne and azide groups can be used as pendants on a polymer chain. Quémener et al. [75] polymerized propargyl methacrylate using cyanoisopropyl dithiobenzoate as a RAFT agent and protecting the pendant acetylene functionality with a silyl group. In parallel, they polymerized vinyl acetate on an azide-functionalized xanthate. The two polymers were clicked together to afford a narrow-dispersed comb copolymer with a *M*_n_ = (3.4–12.5) kg⋅mol^−^^1^ and *Ð* = 1.12–1.18. Later, a similar strategy was put into practice [207] by grafting an alkynyl-terminated polymer onto pendant azide groups of another macromolecule. To that end, glycidyl methacrylate was polymerized in the presence of 4-cyano-4(phenylcarbonothioylthio) pentanoic acid as a RAFT agent and treated with sodium azide and ammonium chloride to functionalize each monomer unit with an azide group. In parallel, the same RAFT agent was coupled with propargyl alcohol and used for the two-step block copolymerization of methyl methacrylate and oligo(ethylene glycol) methyl ether methacrylate. The two obtained polymers were clicked together to form a block copolymer brush (Figure 24) with *M*_n_ = 86 kg⋅mol^−^^1^ and *Ð* = 1.32, which was able to aggregate in selective solvents and thin films due to the hydrophobicity of the polymer backbone and the first grafted block and hydrophilicity of the second grafted block. 

Azide–alkyne cycloaddition can also be used for the preparation of nanocomposites. Islam et al. [208] synthesized a random copolymer of methyl methacrylate and 2-hydroxyethyl methacrylate (70/30 vol.) with *M*_n_ = 16 kg⋅mol^−^^1^ and *Ð* = 1.61 in the presence of 2-cyano-2-propyl dodecyl trithiocarbonate as a RAFT agent. Alkynyl side groups were introduced by esterification between 4-pentanoic acid and side hydroxyl groups of the copolymer. Azide-substituted polyhedral oligosilsesquioxane (POSS) was prepared by the reaction of chloropropyl-heptaisobutyl-substituted POSS with sodium azide. The Huisgen reaction grafted the polymer to POSS and afforded a hybrid nanocomposite (Figure 25), in which a thick layer of polymer brushes was immobilized on the POSS cubic nanostructures, as evidenced by electron microscopy. The inclusion of POSS in the copolymer matrix considerably improved its thermal stability. Later [209], the same idea was implemented with multi-walled carbon nanotubes treated with 4-azidobutylamine. The mass of the grafted polymer layer was nearly equal to the nanotube mass. Polymer-coated nanotubes demonstrated better dispersibility in the polymer matrix. An alternative strategy in the fabrication of nanocomposites is based on the grafting polymerization from inorganic surfaces. Li et al. [210] used silica nanoparticles with an anchored 4-cyanopentanoic acid dithiobenzoate as a RAFT agent to polymerize 6-azidohexyl methacrylate with considerable control over the MMD (*M*_n_ = 25 kg⋅mol^−^^1^ and *Ð* ≤ 1.2) up to 20% conversion. The possibility of subsequent functionalization of polymer-grafted nanoparticles by azide–alkyne cycloaddition with various functional alkynes was explored, including kinetic studies and a comparison with the same reaction on free poly(6-azidohexyl methacrylate). The rate of the click reaction depended on the MM and conversion. On the one hand, grafting to nanoparticles increases the local concentration of reacting polymer side groups, but, on the other hand, there are diffusive limitations at high conversions and high MM of the reactants. It was also demonstrated that attaching the reacting alkynes to oligoanyline increases the conductivity of the covered nanoparticles up to the semiconductor range.

Generation of the azide–alkyne cycloaddition click catalyst Cu(I) in the course of single-electron transfer (SET)-initiated RAFT polymerization provides an interesting opportunity to conduct these processes as tandem reactions. Zhang et al. [211] carried out a one-pot and single-step polymerization of propargyl methacrylate and its click reaction with O-(2-azido-ethyl) S-benzyl dithiocarbonate using ethyl α-bromoisobutyrate as the initiator and 2-cyanoprop-2-yl 1-dithionaphthalate as a RAFT agent. The process kept living characteristics (*Ð* = 1.55, possibility of chain extension with MMA) and proceeded at room temperature. Later, Lu et al. [212] used the same strategy for grafting galactose azide to alkyne side functionality of poly(propargyl methacrylate) chain to obtain a glycopolymer, which was able to adsorb at the surface of gold nanorods after reduction of the polymer end group to thiol. The resulting nanorods can be used for biomolecular recognition. Wang et al. [213] implemented the above one-pot process using 3-(methoxycarbonyl phenylmethylsulfanylthiocarbonylsulfanyl) propionic acid as an iniferter that enabled much faster polymerization rate of methacrylates at room temperature and an easy control over the reacting system by turning on and off the source of visible light (Figure 26). Shen et al. [214] developed a one-pot SET-RAFT-click process for 11-azidoundecanoyl methacrylate, which was polymerized in the presence of 2-cyanoprop-2-yl 1-dithionaphthalatene as a RAFT agent and simultaneously clicked with 4-propargyloxy-40-methoxy azobenzene at room temperature. Depending on the azide–alkyne group ratio in the reacting system, a homopolymer with triazole functional groups (*M*_n_ = 15 kg⋅mol^−^^1^ and *Ð* = 1.19) and a statistical copolymer containing azide and triazole side groups (*M*_n_ = 21 kg⋅mol^−^^1^ and *Ð* ≤ 1.30) were synthesized. 

The use of strained alkynes ensures the azide–alkyne cycloaddition without metal-containing catalysts. Li et al. [215] introduced 5 mol.% of a cyclophenone-masked dibenzoazacyclooctyne monomer in the statistical polymerization of different acrylic monomers with 4-cyano-4-[(dodecylsulfanylthiocarbonyl)sulfanyl]pentanoic acid as a RAFT agent. Polymerizations were performed under visible light (560 nm) using an oxygen tolerant porphyrin-catalyzed photoinduced electron transfer RAFT process, after which the deprotection of the strained alkyne and click reaction with azide-terminated poly(ethylene glycol) were successfully triggered by UV light (290–350 nm). The authors also demonstrated the possibility to incorporate of a second (the same but unprotected) strained alkyne comonomer into the RAFT polymerization of acrylates and to independently click it with the azide moiety prior to the deprotection of the first comonomer.

The hetero Diels–Alder RAFT click concept involving the dienophilic ZCS_2_ group was already discussed in one of the previous sections of this review. Here, we address the situation when a complementary diene belongs to the side group of a RAFT polymer. Bousquet et al. [216] polymerized styrene with a small fraction of diene-containing acrylate units in the presence of trithiocarbonate. The reaction has to be terminated at 20% conversion to prevent crosslinking, which was enough to form 4–5 diene functions per chain with a *M*_n_ ~ 5 kg⋅mol^−^^1^. Then, it was clicked with poly(n-butyl acrylate) of *M*_n_ = (3–13) kg⋅mol^−^^1^, separately synthesized using pyridin-2-yldithioformate as a RAFT agent. Grafting proceeded for 12–24 h with 75–100% efficiency. The authors noted that grafting onto with long chains required their excess relative to side-chain functional groups to reach high yields. Later [217], styrene was replaced with *tert*-butyl acrylate, and diene was introduced into the polymer backbone after RAFT using a two-step procedure. Click grafting with PS, poly(n-butyl acrylate), poly(*tert*-butyl acrylate), and poly(oligoethylene glycol methyl ether acrylate), followed by the hydrolysis of poly(tert-butyl acrylate) into poly(acrylic acid), allowed the authors to obtain a set of hydrophobic, amphiphilic, and hydrophilic comb polymers with narrow MMD. The copolymer with ethylene glycol moiety was able to effectively shield the charges on the surface of hybrid nanoparticles.

Aldehyde–aminooxy click reaction was used by Huang et al. [218] for the preparation of a multi-arm star graft polymer (Figure 27). Styrene and acrolein were copolymerized in the presence of S-1-dodecyl-Sʹ-(α,αʹ-dimethyl-αʹʹ-acetic acid)trithiocarbonate RAFT agent, and star-shape objects with *M*_n_ = 39.8 kg⋅mol^−^^1^ and *Ð* = 1.41 (that corresponded to 21 arm) were obtained via crosslinking of the chain ends with divinylbenzene. The aminooxy-terminated poly(ethylene glycol) with *M*_n_ = 2 kg⋅mol^−^^1^ was then grafted onto aldehyde side groups of the acrolein block with grafting efficiency up to 90% after 24 h. The resulting polymer formed unimolecular micelles in water at high density of grafted poly(ethylene glycol) side chains.

A simple strategy to obtain copolymers with isocyanate groups was implemented by Moraes et al. [219] who added an isocyanate-functionalized monomer at a certain moment of the RAFT polymerization of methyl methacrylate (MMA) or styrene (Figure 28). The product was a MMA-based block copolymer of *M*_n_ = 9.6 kg⋅mol^−^^1^ and *Ð* = 1.61 or a styrene-based block copolymer of *M*_n_ = 5.1 kg⋅mol^−^^1^ and *Ð* = 1.29, with one block built from the non-functionalized monomer (MMA or styrene) and the second being a statistical copolymer of the non-functionalized and functionalized comonomers. Then, a click reaction of side isocyanate groups with hydroxyl groups of phenol, poly(ethylene glycol) methyl ether, or cellulose solid surface was performed at 60 °C, which required up to 60 h for completion. Though the grafting was successful, the resulting polymers could hardly be considered as well-defined systems because of branching at isocyanate groups. The authors noted that that the less reactive (by sterical reasons) functional group used for styrene modification resulted in better results from the viewpoint of less side reactions.

Flores et al. [220] demonstrated the possibility to homopolymerize an unprotected isocyanate-containing monomer, 2-(acryloyloxy)ethylisocyanate, using a neutral RAFT agent, N,N-dimethyl-S-thiobenzoylthiopropionamide, at 50 °C. The resulting polymer was successfully clicked with model low-molar-mass amines, thiols (fast reaction), and alcohols (slow reaction). Later [221], they proposed an inverse approach, where 2-(acryloyloxy)ethylisocyanate or allylisocyanate were used for the click modification of N-(2-hydroxyethyl)acrylamide blocks in the diblock copolymer with N,N-dimethylacrylamide synthesized by RAFT polymerization. The modified block possessed alkene functionality, which was then involved in the second click reaction of the Michael or free radical thiol–ene addition type. A rich variety of thiolating agents allowed the authors to obtain functional block copolymers with potential application in stimuli-responsive systems, bioconjugates, and interpolyelectrolyte complexes.

Thiolation of side groups can be implemented without using isocyanates. Wang et al. [222] subsequently polymerized 3-O-methacryloyl-1,2:5,6-di-O-isopropylidene-D- glucofuranose (*M*_n_ = 7.4 kg⋅mol^−^^1^ and *Ð* = 1.30) and 2-hydroxyethyl methacrylate (*M*_n_ = 8.9 kg⋅mol^−^^1^ and *Ð* = 1.24) using 4,4’-azobis(4-cyanopentanoic acid) as a RAFT agent, removed the thiocarbonylthio end group, converted hydroxyl side groups of the acrylic block into alkenes by esterification with acryloyl chloride, and hydrolyzed the glycopolymer block into poly(3-O-methacryloyl-a,b-D-glucopyranose) in mild conditions. The product was subjected to the thiol–ene click reaction with an endogenic peptide L-glutathione to obtain a peptide–glycopolymer bioconjugate that self-assembled into spherical micelles in water and demonstrated promising pH-sensitive properties, protein recognition, and biodegradability. An even more practical approach to the thiol–ene reaction was developed by Zhang et al. [223] who used S-1-dodecyl-S′ (α,α′-dimethyl-α″-acetic acid) trithiocarbonate RAFT agent to prepare a triblock copolymer, poly(allyl methacrylate)-*b*-poly(ethylene glycol)-*b*-poly(allyl methacrylate), by two-step polymerization. After removal of the thiocarbonylthio end group, the terminal blocks were partially thiolated with an endogenic amino acid L-cysteine, fully thiolated with poly(3-mercaptopropyl)methylsiloxane, and finally crosslinked under UV radiation. The product formed a cysteine-conjugated amphiphilic co-network, which demonstrated optical and mechanical properties suitable for the fabrication of anti-biofouling soft contact lenses. Yang et al. [224] used a thiol–ene reaction to immobilize poly((MMA-*co*-propargyl methacrylate)-*b*-MMA)copolymer with *M*_n_ = (9.4–35.8) kg⋅mol^−^^1^, *Ð* = 1.27-1.54, and 25 mol.% of propargyl units in the corresponding block, obtained by redox-initiated RAFT polymerization, onto attapulgite clay surface treated with γ-mercaptopropyl trimethoxysilane. Then, 2 wt% of the polymer-modified clay was introduced into a polycarbonate matrix to decrease its viscosity and glass transition temperature without compromising mechanical properties and thermal stability. The authors do not explain the nature of this effect but note that it becomes more pronounced with elongation of the MMA block in the copolymer.

He et al. [225] demonstrated that two click reactions in a row, thiol–yne and thiol–ene, can endow polymer side groups with two functionalities (Figure 29). Starting from hydroxyethyl methacrylate, they carried out esterification with propiolic acid and polymerization in the presence of 4-cyano-4-(phenylcarbonothioylthio)pentanoic acid as a RAFT agent to obtain polymer P1 with a low yield (30%) due to crosslinking. The yield can be increased up to 70% if the alkyne moiety is protected with the trimethylsilyl group during polymerization. Then, P1 was clicked with octane-1-thiol as a nucleophile in the presence of triethylamine as an organocatalyst to obtain P2 polymer. In turn, polymer P2 was treated with different thiols in the presence of triazabicyclodecene as an organocatalytic base. Both thiolation steps proceeded with 100% conversion within a few hours. The resulting P3–P6 homopolymers possess functional side groups in each repeating unit. For example, polymer P6 combines hydrophilic tetraethylene glycol and hydrophobic octyl tails and, due to pronounced amphiphilicity, forms micelle-like aggregates in water, which can be used for drug delivery.

Another route to bifunctional copolymers was explored by Oh et al. [226] who synthesized a triblock copolymer, poly[(trimethylsilyl-propargyl methacrylate)-*b*-(triethylene glycol monoethyl ether monomethacrylate-*stat*-glycidyl methacrylate)-*b*-(trimethylsilyl propargyl methacrylate), in three steps with 4-cyano-4-(phenyl carbonothioylthio) pentanoic acid as a RAFT agent. The copolymer with a total *M*_n_ = 23 kg⋅mol^−^^1^ and *Ð* = 1.34 contained 55 protected alkyne group in the terminal blocks and 13 reactive epoxy groups in the middle block. First, epoxy groups were clicked with thiols bearing three types of protecting groups, as shown in the Figure 30, with conversion over 85%. Then, alkyne groups were deprotected and clicked with azides attached to natural sugar D-mannose via the Huisgen reaction with conversion of about 90%. Finally, the thiol groups were deprotected in acid conditions and the obtained copolymers were tested for lectin recognition. With two functionalities, it was possible to compare the contributions from hydrogen bonding and electrostatic interactions and to conclude that the former factor dominates in the binding of a chosen protein.

Thiol–disulfide exchange proceeds as a click reaction upon changing the pH of the medium. This opportunity was explored by Hrsic et al. [227] who synthesized an acrylate or methacrylate-based amphiphilic diblock copolymer in two steps by RAFT in the presence of 4-cyano-4-(phenylcarbonothioylthio) pentanoic acid. The hydrophilic block contained side chains of oligoethylene glycol methyl ether methacrylate with *M*_n_ = 475 g⋅mol^−^^1^ and the hydrophilic block bore acetylthiohexyl pendants. Such copolymers are able to form core–shell micelles in solution. Deprotection of thiol groups via aminolysis, which not only involved side groups but also dithiobenzoate end groups, allowed pH-controlled crosslinking within the hydrophobic cores, thus promoting the formation of well-defined macromolecular structures.

To summarize, it is evident that the field of polymer modification has been substantially enriched by introducing both RAFT polymerization and click reaction concepts. They have significantly revitalized the research on macromolecular reactions in solutions—an area that seemed to be studied far and wide a quarter of century ago [228].

## 6. Conclusions

In this review, we considered only one type of reversible deactivation radical polymerization. This choice is conditioned by the high versatility of RAFT-based polymers from the standpoint of modification opportunities for both chain-end groups and by low sensitivity of the RAFT process to the chemical nature of the functional groups. The interest to the combination of RAFT polymerization and click chemistry is still growing and new publications appear almost every week [229,230,231,232,233].

Whereas azide–alkyne copper-catalyzed cycloaddition undoubtedly remained the most attractive click reaction during the 2000s, the last decade saw an active development of approaches that do not require application of metal-containing catalysts and subsequent purification of polymer products from the catalyst residues, which is especially important for the use of click-functionalized polymers in biology and medicine. Another general tendency regarding click reactions on polymers is that the absence of side reactions is more important than high yield and fast reaction rate.

Many synthetic strategies considered in the present review represent a significant advance in polymer chemistry and design of functional polymers. As a rule, their products are copolymers with chemically different blocks and complex architecture. They have the abilities to form well-defined micelles for carrying drugs and their controlled release, to recognize certain proteins and other biomolecules via selective binding, to localize at the interfaces in multicomponent systems, and to form protective and modifying layers at inorganic surfaces. At the same time, most of the syntheses proposed in the academic literature are too complicated for the development of related industrial technologies. This factor forces researchers to resort to commercially available monomers, one-pot approaches, and tandem or orthogonal reactions in order to increase the practical appeal of their ingenious ideas. We hope that these efforts will bring success within the current decade.

## Data Availability

The data presented in this study are available on request from the corresponding authors.

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
