# Peer review of "RAFT-Based Polymers for Click Reactions"

_polymers, 2022, doi:10.3390/polym14030570_

Round 1

Reviewer 1 Report

Dear all,

Greetings

Please find enclosed our comments regarding this review 

Referenced as : polymers-1575796

Titled: RAFT–based Polymers for Click Reactions

the authors report a review about the reversible addition-fragmentation chain transfert polymerisation plus different types of click reaction, the review enjendre 220 references.

In my opinion this review can be accepted for publication in Polymers journal after fixing all these points ( Minor Revisions).

Title: ok 

Abdtract: cite some potential applications of these polymers

Keywords: cycloaddition include Diels-Alder reactions)

Introduction:  for each criteria such as efficiency , yields, mild conditions, ....add references near to each one.

the synthesis of macromolecules with complex architecture (if you can draw one or two exemples).

Table 1 splite between two pages.

when you presente product I page 6, and II II, IV npage 7, they are figures please add title to them in all your manuscript.

In Table 2: references presentation are not the same in the table 1 and in Table 4?

Table 5 is splited between two pages.

Scheme 25 redrw the structures confusion between CH3 and brakets .

in the conclusion if it's possible to add the industrial uses of these synthetic methodologies.

references pleasqe add some updates from 2021 and 2022

with regards

Author Response

the authors report a review about the reversible addition-fragmentation chain transfert polymerisation plus different types of click reaction, the review enjendre 220 references.

In my opinion this review can be accepted for publication in Polymers journal after fixing all these points ( Minor Revisions).

Title: ok 

Abdtract: cite some potential applications of these polymers

REPLY: Added at the end of Abstract 

Keywords: cycloaddition include Diels-Alder reactions)

REPLY: Corrected

Introduction:  for each criteria such as efficiency , yields, mild conditions, ....add references near to each one.

REPLY: These are criteria necessary for ALL click reactions and it is not clear how to illustrate them with different references. We prefer to classify click reactions by their type in Scheme 1.

the synthesis of macromolecules with complex architecture (if you can draw one or two exemples).

REPLY: We think that enough schemes and figures depicting polymers with complex architecture are presented in the review. It is not clear why extra examples are needed in the introduction.

Table 1 splite between two pages.

REPLY: Corrected

when you presente product I page 6, and II II, IV npage 7, they are figures please add title to them in all your manuscript.

REPLY: All previously untitled chemical structures were made figures and added with captions.

In Table 2: references presentation are not the same in the table 1 and in Table 4?

REPLY: Corrected

Table 5 is splited between two pages.

REPLY: Corrected

Scheme 25 redrw the structures confusion between CH3 and brakets .

REPLY: Corrected

in the conclusion if it's possible to add the industrial uses of these synthetic methodologies.

REPLY: Conclusions section is expanded to add potential uses of the proposed synthetic approaches

references pleasqe add some updates from 2021 and 2022

REPLY: Some fresh papers were added as refs 229-233

Reviewer 2 Report

Elena V. Chernikova and Yaroslav V. Kudryavtsev fully reviewed RAFT–based Polymers for Click Reactions. They focused on the main topic and described it very carefully in this manuscript. It’s a very useful and fantastic review for the introduction of the RAFT technique used in click reactions. The reviewer considered that this work is valuable to be published on Polymers but after a few revisions below.

In this manuscript, some places need to be improved and carefully checked, such as:

Page 3. Line 73, the term “ reversible deactivation radical polymerization (RDRP)” should be changed to “reversible-deactivation radical polymerization(RDRP)” and insert the citations “Jenkins, A. D.; Jones, R. G.; Moad, G., T. IUPAC Recommendations 2010. Pure and Applied Chemistry 2009, 82 (2), 483-491”.

In addition, for the sake of comparison, not only RAFT, the author but also should better add some sentences to describe the characteristics of NMP and ATRP at the end of this paragraph. Matyjaszewski, K group has rich experience and does many excellent outcomings. Especially, the application of the electrochemistry technique induced the rapid development of eRAFT and eATRP. (Please the author inserts these citations: Corrigan, N.; Jung, K.; Matyjaszewski, K. et al Progress in Polymer Science 2020, 101311; Wang, Y.; Matyjaszewski, K. et al, Macromolecules 2017, 50 (20), 7872-7879; Luo, J.; Gennaro, A.; et al, Electrochimica Acta 2021, 388, 138589; Gennaro, A.; Matyjaszewski, K. et al, Science 2011, 332 (6025), 81-4). Other skills like photo-polymerization, mechanical-polymerization should be also mentioned.

Page 6, line 156, please author insert several sentences and list some specific examples that used click polymerization, like zero dimension material: quantum dots, Al2O3/SiO2/Fe2O3 sphere material etc; 1D materials: carbon nanotube (CNT), nano rod etc; 2 D materials: graphene, BN, MoS2 etc.

Page 8, line 224, SEC is an abbreviation word, please complete the full name

Page 9, line 251, “…is proposed in [95],” to “…is proposed in Ref. [95],”, like line 405

       Line 254, “ as well of…” to “ as well as…”

In the 2nd partial, authors carefully introduced the diverse R and Z groups how to affect the click reaction. However, most of the paragraphs just list some special cases but not described the mechanism of real reason that caused this difference. Actually, the Z group affects the reactivity of the C=X bond and the stability of the intermediate radical, the generated new radical R• is capable of reacting with the monomer to generate a new active radical chain. Thus, is it possible to make up several sentences to conclude the important function of different structures of R and Z groups? For instance, try to describe from the two viewpoints of the steric and electronic factors.

Page 10, line 287, please insert several citations

Line 302, GPC is an abbreviation word, please complete the full name

Line 317, “behavior” to the plural noun “behaviors”

Page 12, line 354, “…was realized in [155]” is revised to “…was realized in Ref. [155]”, like line 405

Page 13, Line 374, try to expand the description (maybe one or two sentences) of the doubts and the reasons of Zhou et. al in Ref. [165].

Page 17, Table 6, Please author check the term “VAa)”.

Page 18, line 531, the term of “described in [35]”

Page 28, in the final conclusions part, the author honestly pointed out the biggest disadvantage of organic compounds synthetic strategies. This is a quite difficult issue for all chemists, especially for organic chemistry. However, it’s necessary for the development of drugs, biology, anti-corrosion, modifying nano materials and some other products. Thus, the author also has to face the real problem of RAFT based polymers for click polymerizations. Not only the research outcomes but also the issues have to be concluded. In addition, the prospect of this area in the future should be broadened as well. Please enrich this paragraph.

Author Response

Elena V. Chernikova and Yaroslav V. Kudryavtsev fully reviewed RAFT–based Polymers for Click Reactions. They focused on the main topic and described it very carefully in this manuscript. It’s a very useful and fantastic review for the introduction of the RAFT technique used in click reactions. The reviewer considered that this work is valuable to be published on Polymers but after a few revisions below.

In this manuscript, some places need to be improved and carefully checked, such as:

Page 3. Line 73, the term “ reversible deactivation radical polymerization (RDRP)” should be changed to “reversible-deactivation radical polymerization(RDRP)”

REPLY: Corrected

and insert the citations “Jenkins, A. D.; Jones, R. G.; Moad, G., T. IUPAC Recommendations 2010. Pure and Applied Chemistry 2009, 82 (2), 483-491”.

REPLY: Inserted as ref. 39

In addition, for the sake of comparison, not only RAFT, the author but also should better add some sentences to describe the characteristics of NMP and ATRP at the end of this paragraph. Matyjaszewski, K group has rich experience and does many excellent outcomings. Especially, the application of the electrochemistry technique induced the rapid development of eRAFT and eATRP. (Please the author inserts these citations: Corrigan, N.; Jung, K.; Matyjaszewski, K. et al Progress in Polymer Science 2020, 101311; Wang, Y.; Matyjaszewski, K. et al, Macromolecules 2017, 50 (20), 7872-7879; Luo, J.; Gennaro, A.; et al, Electrochimica Acta 2021, 388, 138589; Gennaro, A.; Matyjaszewski, K. et al, Science 2011, 332 (6025), 81-4). Other skills like photo-polymerization, mechanical-polymerization should be also mentioned.

REPLY: We are aware of NMP and ATRP techniques, as well as of achievements by the Matyjaszewski group, but a meaningful comparison between different RDRP approaches would require another chapter in the review.

Page 6, line 156, please author insert several sentences and list some specific examples that used click polymerization, like zero dimension material: quantum dots, Al2O3/SiO2/Fe2O3 sphere material etc; 1D materials: carbon nanotube (CNT), nano rod etc; 2 D materials: graphene, BN, MoS2 etc.

REPLY: Our review is not about click polymerization, it is about the combination of RAFT polymerization and click modification

Page 8, line 224, SEC is an abbreviation word, please complete the full name

REPLY: Corrected

Page 9, line 251, “…is proposed in [95],” to “…is proposed in Ref. [95],”, like line 405

REPLY: Corrected

       Line 254, “ as well of…” to “ as well as…”

REPLY: Corrected

In the 2nd partial, authors carefully introduced the diverse R and Z groups how to affect the click reaction. However, most of the paragraphs just list some special cases but not described the mechanism of real reason that caused this difference. Actually, the Z group affects the reactivity of the C=X bond and the stability of the intermediate radical, the generated new radical R• is capable of reacting with the monomer to generate a new active radical chain. Thus, is it possible to make up several sentences to conclude the important function of different structures of R and Z groups? For instance, try to describe from the two viewpoints of the steric and electronic factors.

REPLY: A new  paragraph is inserted on page 3 next to Scheme 2 to specify the difference between the roles of the R and Z groups in RAFT polymerization. At the same time we prefer not to discuss the atomic details of the mechanism of the process as this is of low interest for the majority of polymer chemists.

Page 10, line 287, please insert several citations

REPLY: Corrected

Line 302, GPC is an abbreviation word, please complete the full name

REPLY: Corrected

Line 317, “behavior” to the plural noun “behaviors”

REPLY: Corrected

Page 12, line 354, “…was realized in [155]” is revised to “…was realized in Ref. [155]”, like line 405

REPLY: Corrected

Page 13, Line 374, try to expand the description (maybe one or two sentences) of the doubts and the reasons of Zhou et. al in Ref. [165].

REPLY: Expanded on page 13

Page 17, Table 6, Please author check the term “VAa)”.

REPLY: Corrected

Page 18, line 531, the term of “described in [35]”

REPLY: Corrected

Page 28, in the final conclusions part, the author honestly pointed out the biggest disadvantage of organic compounds synthetic strategies. This is a quite difficult issue for all chemists, especially for organic chemistry. However, it’s necessary for the development of drugs, biology, anti-corrosion, modifying nano materials and some other products. Thus, the author also has to face the real problem of RAFT based polymers for click polymerizations. Not only the research outcomes but also the issues have to be concluded. In addition, the prospect of this area in the future should be broadened as well. Please enrich this paragraph.

REPLY: The Conclusions part is expanded to mention possible industrial applications and to make the paper end generally less pessimistic.